# Comparison of Bioactive Phenolic Compounds and Antioxidant Activities of Different Parts of *Taraxacum mongolicum*

**DOI:** 10.3390/molecules25143260

**Published:** 2020-07-17

**Authors:** Li Duan, Chenmeng Zhang, Yang Zhao, Yanzhong Chang, Long Guo

**Affiliations:** 1College of Chemistry and Material Science, Hebei Normal University, Shijiazhuang 050024, China; duanli@hebtu.edu.cn (L.D.); zhangchenmeng08@163.com (C.Z.); zhaoyang7613@163.com (Y.Z.); 2Laboratory of Molecular Iron Metabolism, College of Life Science, Hebei Normal University, Shijiazhuang 050024, China; chang7676@163.com; 3Traditional Chinese Medicine Processing Technology Innovation Center of Hebei Province, Hebei University of Chinese Medicine, Shijiazhuang 050200, China

**Keywords:** *Taraxacum mongolicum*, phenolics, antioxidant activity, multivariate statistical analysis

## Abstract

Herbs derived from *Taraxacum* genus have been used as traditional medicines and food supplements in China for hundreds of years. *Taraxacum mongolicum* is a famous traditional Chinese medicine derived from *Taraxacum* genus for the treatment of inflammatory disorders and viral infectious diseases. In the present study, the bioactive phenolic chemical profiles and antioxidant activities of flowers, leaves, and roots of *Taraxacum mongolicum* were investigated. Firstly, a high performance liquid chromatography method combined with segmental monitoring strategy was employed to simultaneously determine six bioactive phenolic compounds in *Taraxacum mongolicum* samples. Moreover, multivariate statistical analysis, including hierarchical clustering analysis, principal component analysis, and partial least squares discriminant analysis were performed to compare and discriminate different parts of *Taraxacum mongolicum* based on the quantitative data. The results showed that three phenolic compounds, caftaric acid, caffeic acid, and luteolin, could be regarded as chemical markers for the differences of flowers, leaves, and roots of *Taraxacum mongolicum*. In parallel, total phenolic contents, total flavonoid contents and antioxidant activities of different parts of *Taraxacum mongolicum* were also evaluated and compared. It is clear that *Taraxacum mongolicum* had antioxidant properties, and the antioxidant capacities of different parts of *Taraxacum mongolicum* in three antioxidant assays showed a similar tendency: Flowers ≈ leaves > roots, which revealed a positive relationship with their total phenolic and flavonoid contents. Furthermore, to find the potential antioxidant components of *Taraxacum mongolicum*, the latent relationships of the six bioactive phenolic compounds and antioxidant activities of *Taraxacum mongolicum* were investigated by Pearson correlation analysis. The results indicated caftaric acid and caffeic acid could be the potential antioxidant ingredients of *Taraxacum mongolicum*. The present work may facilitate better understanding of differences of bioactive phenolic constituents and antioxidant activities of different parts of *Taraxacum mongolicum* and provide useful information for utilization of this herbal medicine.

## 1. Introduction

The *Taraxacum* genus is a member of the Asteraceae family, and widely distributed in the warmer temperature zones of the northern hemisphere [1,2]. In China, the *Taraxacum* genus includes more than 70 species, and herbs derived from *Taraxacum* genus have been used as traditional medicines and food supplements since the ancient time [3]. *Taraxacum mongolicum* (TM), a famous medicinal herb, has been reported to possess antioxidant, anti-inflammatory, anti-cancer, anti-hyperglycemic, anti-coagulatory, and analgesic activities, and commonly used for treatment of inflammatory disorders and viral infectious diseases [4,5,6]. These health-promoting effects may be attributed to the presence of bioactive compounds in TM. Furthermore, the leaves of TM are also consumed as vegetable food, and the extract is used as flavour ingredients in various food products, such as alcoholic beverages, soft drink and baked goods [7].

Phytochemical researches demonstrated that TM contains various bioactive compounds, such as phenolics, flavonoids, terpenes and coumarins. Among bioactive constituents, phenolics and flavonoids are the most abundant compounds and also recognized as the main active compounds in TM [8,9]. Although TM is used as herbal medicines and food ingredients for a long time, studies on its bioactive compositions are still limited. To date, several analytical methods, including high performance liquid chromatography (HPLC) and high performance liquid chromatography coupled with mass spectrometry (HPLC-MS) have been used for qualitative and quantitative analysis of main bioactive components in TM, but most of the published works focused on whole herb, the bioactive chemical compounds of different parts (flowers, leaves, and roots) of TM have never been researched and compared [10,11,12].

The antioxidative activities of herbal medicines have received a great amount of attention as being primary preventive ingredients against various diseases, such as cancer, inflammation, and cardiovascular diseases [13]. Recent studies have described that the antioxidant properties of herbal medicines are mainly due to their phenolic compounds, such as flavonoids and phenolic acids [14,15]. Although the preliminary studies showed that the TM extract exhibited high antioxidative activity, no research has been conducted to compare the antioxidant activities of different parts of TM, and the associations between the bioactive compounds and the antioxidative activity of TM was also unclear.

To evaluate and compare the phenolic compositions of different parts of TM, and find the potential antioxidant components of TM, in this study, an HPLC method combined with segmental monitoring strategy was established to quantitate six bioactive constituents (caftaric acid, chlorogenic acid, caffeic acid, cichoric acid, 3,5-di-*O*-caffeoylquinic acid, and luteolin) in different parts (flowers, leaves, and roots) of TM. Then, the quantitative results were analyzed by multivariate statistical analysis, including hierarchical clustering analysis (HCA), principal component analysis (PCA) and partial least squares discriminant analysis (PLS-DA) to compare and distinguish the different parts of TM and find the chemical makers. Additionally, the total phenolic, total flavonoid, and antioxidant activities of different parts of TM were evaluated, and the Pearson correlation analysis was carried out to investigate the latent relationships of the bioactive compounds and antioxidant activities of TM samples.

## 2. Results and Discussion

### 2.1. Comparison of Six Bioactive Phenolic Compounds in Different Parts of Taraxacum mongolicum

#### 2.1.1. Optimization of Extraction Conditions

To get a better extraction efficiency of the six compounds (caftaric acid, chlorogenic acid, caffeic acid, cichoric acid, 3,5-di-*O*-caffeoylquinic acid, and luteolin) of TM, the extraction conditions, including extraction solvents (ethanol, 40% methanol, 60% methanol, 80% methanol, and 100% methanol), extraction time (30, 40, 50, and 90 min) and liquid-solid ratios (25 mL/g, 50 mL/g, and 100 mL/g) were optimized using univariate test. The results showed that 60% methanol was the most efficient extraction solvent. Finally, the optimal extraction conditions of six phenolic components were extraction solvent, 60% methanol; extraction time, 40 min; liquid-solid ratio, 50 mL/g.

#### 2.1.2. Method Validation of HPLC Analysis

The proposed HPLC method was validated by determination of linearity, limit of detections (LOD), limit of quantifications (LOQ), precision, stability, repeatability and recovery. Acceptable linear correlations of the calibration curves were obtained (r^2^ ≥ 0.9998) within the test range. The LOD were less than 0.043 mg/L, and the LOQ were less than 0.188 mg/L. The relative standard deviations (RSD) of the intra- and inter-day precisions were less than 1.22% and 1.68%, respectively. The repeatability presented as RSD were in the range from 0.45% to 1.76%, and the stability were less than 1.76%. The recoveries of the six compounds were in the range of 93.76–108.73% with the RSD less than 2.97%. The results of method validation indicated that the established HPLC method was suitable for the quantitative analysis (Table 1).

#### 2.1.3. Determination of Six Bioactive Phenolic Compounds in Different Parts of *Taraxacum mongolicum*

The developed HPLC method was subsequently applied to quantitatively analysis the flowers, leaves, and roots of TM samples, including 12 batches of TM flowers (F1–F12), 11 batches of TM leaves (L1–L11), and nine batches of TM roots (R1–R9). A total of six phenolic compounds, including caftaric acid, chlorogenic acid, caffeic acid, cichoric acid, 3,5-di-*O*-caffeoylquinic acid, and luteolin were assayed simultaneously. The caftaric acid, chlorogenic acid, caffeic acid, cichoric acid, and 3,5-di-*O*-caffeoylquinic acid showed distinct maximal UV absorption at 325 nm, the luteolin had maximal absorptions at 350 nm. To analyze the six components at one run, the chromatogram was divided into two segments according to the retention time of the target compounds, and set specific detection wavelength for each segment. Therefore, the UV detector was monitored at 325 nm from 0 to 40 min and 350 nm from 40 to 55 min.The typical chromatograms of flowers, leaves and roots of TM are shown in Figure 1, and quantitative results are summarized in Table 2. Obviously, the contents of phenolic compounds in different parts of TM were unstable. Among the six phenolic compounds, chicoric acid was the most abundant constituent, and the average contents were 2.941, 4.341, and 3.686 mg/g in flowers, leaves, and roots, respectively. The TM flowers had relatively high contents of caffeic acid and luteolin (0.456 and 1.113 mg/g on average), while the TM leaves had relatively high contents of caftaric acid (1.400 mg/g on average). It was worth noting that the TM root had relatively low contents of caftaric acid, caffeic acid, and luteolin compared with other two parts of TM. It has been widely reported that bioactive compounds of TM were phenolics and flavonoids. Up to now, several studies have been investigated the bioactive phenolics constitutes in whole herb of TM, but no research has been conducted to compare the contents of phenolic components in different parts of TM [16]. In the present work, the contents of the six phenolic compounds in flowers, leaves and roots of TM have been determined and the results showed that the different parts (flowers, leaves and roots) of TM showed similar chemical profiles, but the content levels of the bioactive compounds varied significantly.

#### 2.1.4. Multivariate Statistical Analysis

To compare and provide more information about the chemical differences of flowers, leaves, and roots of TM, multivariate statistical analysis including HCA, PCA, and PLS-DA were further performed based on the characteristics of the contents of six phenolic compounds.

HCA is a clustering method which explores the organization of samples in groups and among groups depicting a hierarchy. Firstly, HCA was performed using Ward’s method as the cluster method. The HCA dendrogram is demonstrated in Figure 2a. The TM samples were successfully classified into three groups corresponding to flowers, leaves and roots of TM, which indicated that different parts of TM were indeed different in terms of levels of phenolic compounds.

PCA and PLS-DA are mathematical approaches that can be applied to chemical or biological data, in order to recognize pattern and classify the samples, and the two methods have been widely used in discriminating and comparing the composition of herbal medicines [17,18,19]. To provide more information about the chemical differences and find potential chemical markers which contributed to the differences among flowers, leaves, and roots of TM, PCA and PLS-DA were further carried out. Unsupervised PCA was carried out to visualize the classification trends of TM samples based on the contents of the six phenolic compounds. The first and second principal components described 59.4% and 27.1% of the variability in the original observations, and the first two principal components accounted for 86.5% of total variance. As shown in PCA score scatter plot (Figure 2b), the PCA result was similar with HCA, 32 batches of TM samples were classified into three groups. Compared with TM leaves and TM roots samples, the distribution of TM flowers samples is more scattered, which suggested that the quality of TM flowers was less stable.

In order to further find potential chemical markers of flowers, leaves, and roots of TM, supervised PLS-DA was further employed based on the contents of the six phenolic compounds. Similar to the PCA result, the PLS-DA scores plot (Figure 2c) showed that flowers, leaves and roots of TM samples could also be readily classified into three groups with the R2Y = 0.763 and Q2 = 0.708, revealing a good classification and prediction ability of the model. To select the potential chemical markers, the variable importance in projection (VIP) values of the six compounds were calculated. The VIP values represent the differences of the variables, and compounds that played important roles in differentiation were picked out when the VIP values were more than 1.0. Finally, three phenolic compounds, caftaric acid, caffeic acid, and luteolin with their VIP values more than 1.0 were selected as chemical markers, which were responsible for the significantly intergroup differences of different parts of TM samples.

### 2.2. Comparison of Six Bioactive Phenolic Compounds in Different Parts of Taraxacum mongolicum

Phenolic and flavonoid compounds are considered as the most important antioxidative components of herbal medicines and other plant materials. In present study, total flavonoid contents and total phenolic contents of flowers, leaves and roots of TM were determined and compared. As shown in Figure 3a, the total flavonoid contents of TM samples were in the range of 21.29–33.83 mg RE/g (milligrams of rutin equivalent per gram dry weight of sample), and the total flavonoid contents of leaves and flowers were significantly higher than those of roots. The total phenolic contents of different parts of TM samples ranged from 37.12 to 68.89 mg GAE/g (milligrams of gallic acid equivalents per gram dry weight of sample), and the flowers contained relatively high content of phenolic, following leaves and roots (Figure 3b). 

The antioxidant activities of different parts of TM were also determined by three in vitro assays (DPPH, ABTS, and FRAP), which are recommended as rapid, simple, low-cost and reproducible tools for measuring the antioxidant potential of plant extracts. The average antioxidant activities of flowers, leaves, and roots of TM are shown in Figure 3c–e. It is clear that the flowers, leaves, and roots of TM showed antioxidant capacities, and the antioxidant activities of different parts of TM in three antioxidant assays showed a similar tendency: Flowers ≈ leaves > roots. Phenolic and flavonoid compounds were usually considered as the basis of antioxidant activities due to their hydroxyl group [20,21]. Similarly, our results indicated that the antioxidant activities of different parts of TM samples revealed a positive relationship with their total phenolic and flavonoid contents. The flowers and leaves of TM had higher contents of phenolic and flavonoid, they also showed better antioxidant activities.

### 2.3. The Potential Antioxidant Constitutes of Taraxacum mongolicum

The preliminary studies showed that the TM exhibited high antioxidant activity, but the potential antioxidant ingredients were still unknown [22]. To investigate the latent relationships of the six bioactive phenolic compounds and antioxidant activities, and find the potential antioxidant components of TM, the Pearson correlation coefficients were calculated based on the quantities of the contents of six phenolic compounds and the antioxidant activities of DPPH, FRAP, and ABTS assays. The heatmap of Pearson correlation analysis is shown in Figure 4a. It was clear that caftaric acid and caffeic acid had high correlation with the antioxidant activities of TM in three in vitro assays, and these two phenolic compounds could be recognized as mainly antioxidant constitutes of TM. 

To further confirm the Pearson correlation analysis result, the antioxidant activities of six phenolic compounds were determined by DPPH assay. As shown in Figure 4b, the half maximal inhibitory concentration (IC50) of the six phenolic components were 7.31, 12.74, 7.12, 9.55, 10.13, 5.25 μg/mL for caftaric acid, chlorogenic acid, caffeic acid, cichoric acid, 3,5-di-*O*-caffeoylquinic acid, and luteolin, respectively. Among the six phenolic compounds, caftaric acid, caffeic acid, and luteolin showed relative higher antioxidant capacities compared to other three components, which was highly consistent with Pearson coefficient analysis results (DPPH). Therefore, combining the results of Pearson correlation analysis and antioxidant assays, it could be speculated that caftaric acid and caffeic acid were the most potential antioxidant ingredients of TM.

## 3. Materials and Methods

### 3.1. Plant Materials

Twelve batches of TM flowers (F1–F12), eleven batches of TM leaves (L1–L11) and nine batches of TM roots (R1–R9) were purchased in a Chinese herbal medicine market (Anguo, Hebei, China). All the TM samples were dry in the shade and stored in the desiccator (Haier Smart Home Co., Ltd., Qingdao, China.). The plant materials are identified by Associate Professor Long Guo, and the voucher specimens have been deposited in Hebei University of Chinese Medicine, Shijiazhuang, China.

### 3.2. Chemicals and Reagents

Six reference compounds, including caftaric acid (S1), chlorogenic acid (S2), caffeic acid (S3), cichoric acid (S4), 3,5-di-*O*-caffeoylquinic acid (S5), and luteolin (S6) were purchased from Chengdu Must Biotechnology Co., Ltd. (Chengdu, China). The structures of the six reference compounds are shown in Figure 5. The purities of these compounds were determined to be higher than 98% by high performance liquid chromatography diode array detection analysis. Total antioxidant capacity assay kits (ABTS and FRAP method) were purchased from Beyotime Institute of Biotechnology (Shanghai, China). 2,2-Diphenyl-1-picrylhydrazyl (DPPH) and Folin-Ciocalteu reagent were purchased from Macklin (Shanghai Macklin Biochemical Co., Ltd., Shanghai, China). (±)-6-Hydroxy-2,5,7,8-tetramethylchromane-2-carboxylic acid (Trolox) was purchased from Aladdin (Shanghai Aladdin Bio-Chem Technology Co., LTD, Shanghai, China). Gallic acid and rutin were purchased from Chengdu Must Biotechnology Co., Ltd. (Chengdu, China). 

Methanol, acetonitrile and formic acid of HPLC grade were purchased from Thermo Fisher Scientific Inc. (Waltham, MA, USA). Deionized water was prepared by a Milli-Q water purification system (Millipore, Billerica, MA, USA). All other reagents and chemicals used in the experiment were of analytical grade. 

### 3.3. Preparation of Sample and Standard Solutions

All TM samples (flowers, leaves and roots) were powdered and screened through 40 mesh sieves. Each sample (0.5 g) was thoroughly mixed with 60% methanol (25 mL) and extracted by ultrasonator for 40 min at 40 kHz. The extracted solution was centrifuged at 12,000 g for 10 min at room temperature. Then 10 μL of the supernatant was injected into HPLC instrument for analysis.

Standard stock solutions of caftaric acid, chlorogenic acid, caffeic acid, cichoric acid, 3,5-di-*O*-caffeoylquinic acid, and luteolin were prepared by accurately weighing and dissolving in methanol, with the concentrations of 0.68, 0.96, 0.69, 0.87, 0.90, and 0.71 mg/mL, respectively. Prior to analysis, standard stock solutions were appropriately diluted with methanol to a series of appropriate concentrations used as working standard solutions. 

### 3.4. HPLC Analysis

Analysis was performed on an Agilent 1260 Infinity HPLC system (Agilent Technologies, Santa Clara, CA, USA) using Thermo Hypersil BDS C18 column (250 mm × 4.6 mm, 5 μm). The mobile phase consisted of 0.1% aqueous formic acid (A) and acetonitrile (B) using a gradient program of 6–7% B in 0–10 min, 7–10% B in 10–13 min, 10–15% B in 13–17 min, 15–15% B in 17–21 min, 15–18% B in 21–26 min, 18–27% B in 26–35 min, 27–40% B in 35–45 min, 40–60% B in 45–50 min and 60–80% B in 50–55 min. The flow rate was maintained at 1.0 mL/min and the column temperature was maintained at 25 °C. The detection wavelengths were set as follows: 325 nm from 0 to 40 min, 350 nm from 40 to 55 min.

### 3.5. Method Validation of HPLC Analysis

For the calibration curves, six different concentrations of working standard solutions were analyzed. The calibration curves were calculated by plotting the peak areas of each analyte versus its concentrations. The limit of detections (LOD) and limit of quantifications (LOQ) for each analyte were defined by the concentrations that generated peaks with signal-to-noise values (S/N) of 3 and 10, respectively.

The intra-day precision of the developed method was investigated by repetitively analyzing of the sample solution six times within the same day. While for inter-day precision, the sample were examined in triplicates for consecutive three days. For the stability test, the same sample was stored at room temperature and analyzed by replicate injection at 0, 2, 4, 6, 8, 10, 12, 24, and 48 h. To confirm the repeatability, six replicates of the same samples were extracted and analyzed independently. The relative standard deviations (RSD) were used to evaluate precision, stability and repeatability of the established method. Recovery of analytes was carried out for accuracy evaluation of the method. A certain amount of the six standards were added into a 0.25 g powder of six batches of the same sample, and then extracted and analyzed in sextuplicate with the same procedures. The average recoveries were estimated by the formula:
Recovery (%) = (Detected amount − Original)/Spiked amount × 100(1)


### 3.6. Determination of Total Phenolic Contents

The total phenolic contents of different parts of TM were estimated with Folin—Ciocalteu reagent (FC) [23]. Briefly, 250 μL of TM extract was added to 626 μL FC reagent (0.2 mg/mL) and then mixed thoroughly. After 5 min, 1 mL of 7.5% Na_2_CO_3_ solution and 1 mL distilled water were added. Then, the mixture was incubated in the dark at room temperature (25 °C) for 40 min. The absorbance of the mixture was recorded at 727 nm against the blank (60% methanol). The total phenolic contents of TM samples were expressed in milligrams of gallic acid equivalents per gram dry weight of sample (mg GAE/g).

### 3.7. Determination of Total Flavonoid Contents

The total flavonoid contents of different parts of TM were determined based on the method described by Jia et al. with some modifications [24]. In brief, 500 μL of TM extracts and 400 μL of 5% NaNO_2_ were added to a 10 mL volumetric flask. The mixture was left to settle for 6 min. Then, 400 μL of 10% AlCl_3_ was added. After 6 min of reaction, 4 mL of 4% NaOH were added and the total volume was adjusted to 10 mL with distilled. The sample was thoroughly mixed and kept for 15 min at room temperature (25 °C). Then, the absorbance of the mixture was taken at 507 nm. Here, the total flavonoid contents were estimated as milligrams of rutin equivalent per gram dry weight of sample (mg RE/g).

### 3.8. Antioxidant Assays

#### 3.8.1. DPPH Scavenging Activity Assay

DPPH molecule is a stable synthetic free radical which is widely used to evaluate the ability of compounds to act as free radical scavengers or radical hydrogen donors, and the DPPH scavenging activity assay was performed in accordance with a modified method of Tuberoso et al. [25,26]. Briefly, 100 μL of different parts (flowers, leaves, roots) of TM extract (60% methanol) was mixed with 100 μL of DPPH (0.2 mM) in methanol, after incubation for 30 min in darkness. The absorbance was monitored at 519 nm using a microplate reader against a blank (phosphate buffer saline). The decrease in absorbance was proportional to the antioxidant capacity and can be measured in comparison to Trolox as standard. The results of DPPH scavenging activity were expressed as μM Trolox equivalents per gram dry weight of sample (μM TE/g).

#### 3.8.2. ABTS Radical Scavenging Activity Assay

The ABTS radical scavenging activity of different parts of TM was performed by a commercial kit following the instruction manual [27,28]. Ten microliters of different parts (flowers, leaves, roots) of TM extract (60% methanol) was mixed with 190 μL of ABTS reagent. After incubation at room temperature for 6 min, the absorbance was monitored at 414 nm using a microplate reader against a blank (phosphate buffer saline). The antioxidant capacity can be measured in comparison to Trolox as standard. The results of ABTS scavenging capacity were expressed as μM Trolox equivalent per gram dry weight of sample (μM TE/g).

#### 3.8.3. Ferric Reducing Antioxidant Power (FRAP) Assay

The FRAP assay was carried out by a commercial kit following the instruction manual. Under acidic conditions, antioxidants can restore ferric-2,4,6-tris(pyridin-2-yl)-1,3,5-triazine (Fe^3+^-TPTZ) to ferrous complex [29,30]. Five microliters of different parts (flowers, leaves, roots) of TM extract (60% methanol) was mixed with 180 μL of FRAP reagent. After incubation at 37 °C for 4 min, the absorbance of the reaction mixture was measured at 593 nm using a microplate reader. The antioxidant capacity can be measured in comparison to Trolox as standard. The results of FRAP assay were expressed as μM FeSO_4_ equivalent per gram dry weight of sample (μM FeSO_4_/g).

#### 3.8.4. Antioxidant Activities of Six Bioactive Phenolic Compounds

The antioxidant activities of six bioactive phenolic constitutes, including caftaric acid, chlorogenic acid, caffeic acid, cichoric acid, luteolin, and 3,5-di-*O*-caffeoylquinic acid were determined by DPPH scavenging activity assay. As mentioned above, 100 μL of different concentrations of six bioactive phenolic compounds were incubated with DPPH (100 μL, 0.2 mM) in methanol for 30 min in darkness. Then, the absorbance was monitored at 519 nm against a blank (60% methanol). DPPH inhibitions of six bioactive phenolic constitutes were calculated as follows:
DPPH inhibitions (%) = (A_control_ − A_sample_)/A_control_ × 100(2)
where A_control_ is the absorbance values of 60% methanol + 0.2 mM DPPH in methanol, and A_sample_ is the absorbance values of the test samples in 60% methanol + 0.2 mM DPPH in methanol. Based on the values of inhibition and concentration, the IC50 value (the half maximal inhibitory concentration) of each component was calculated.

### 3.9. Data Analysis

Hierarchical clustering analysis (HCA) and Pearson correlation analysis were performed by SPSS 18.0 (SPSS Inc, Chicago, IL, USA). Principal component analysis (PCA) and partial least squares discriminant analysis (PLS-DA) were carried out by SIMCA-P+ 13.0 Software (Umetric, Umea, Sweden). The contents of the six phenolic compounds in flowers, leaves and roots of TM samples were used as input data for HCA, PCA, and PLS-DA.

## 4. Conclusions

In order to evaluate and compare the bioactive phenolic compounds and antioxidant activities of different parts (flowers, leaves, and roots) of TM, in this work, an HPLC method combined with segmental monitoring strategy was firstly applied to quantify the six bioactive phenolic constitutes, including caftaric acid, chlorogenic acid, caffeic acid, cichoric acid, 3,5-di-*O*-caffeoylquinic acid, and luteolin in flowers, leaves, and roots of TM. Furthermore, multivariate statistical analysis, including HCA, PCA, and PLS-DA were performed to provide more information about the chemical differences utilization of TM.

## Figures and Tables

**Figure 1 molecules-25-03260-f001:**
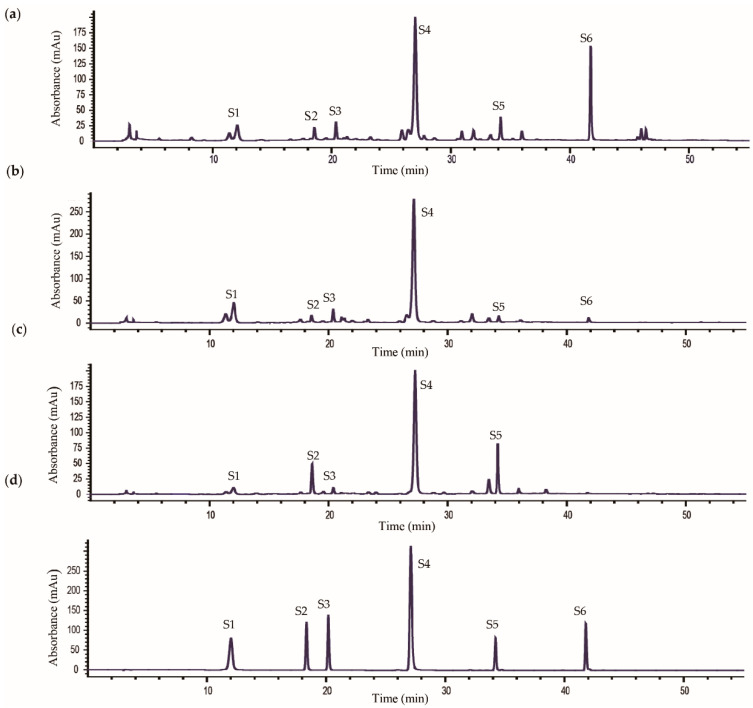
The typical high performance liquid chromatography (HPLC) chromatograms of flowers (**a**), leaves (**b**), roots (**c**) of *Taraxacum mongolicum,* and mixed standard solution (**d**). S1, caftaric acid; S2, chlorogenic acid; S3, caffeic acid; S4, cichoric acid; S5, 3,5-di-*O*-caffeoylquinic acid; S6, luteolin.

**Figure 2 molecules-25-03260-f002:**
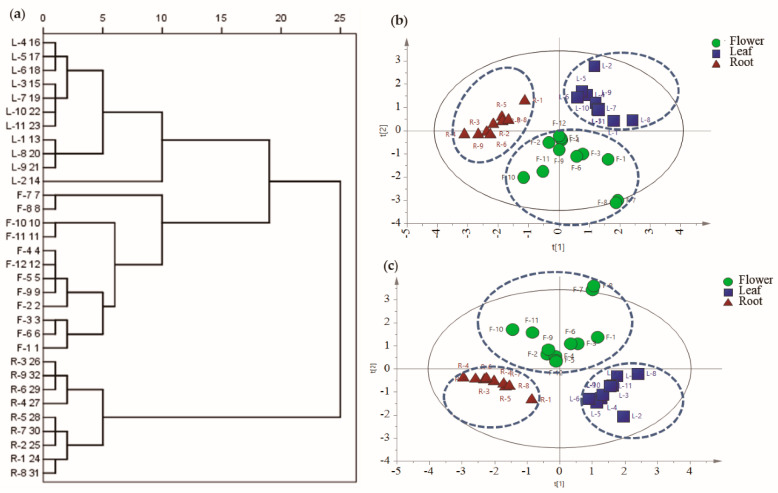
The dendrograms of hierarchical clustering analysis (HCA) (**a**), and the score plots of principal component analysis (PCA) (**b**) and partial least squares discriminant analysis (PLS-DA) (**c**). Green circles represent *Taraxacum mongolicum* flowers (F1–F12); blue squares represent *Taraxacum mongolicum* leaves (L1–L11); red triangles represent *Taraxacum mongolicum* roots (R1–R9).

**Figure 3 molecules-25-03260-f003:**
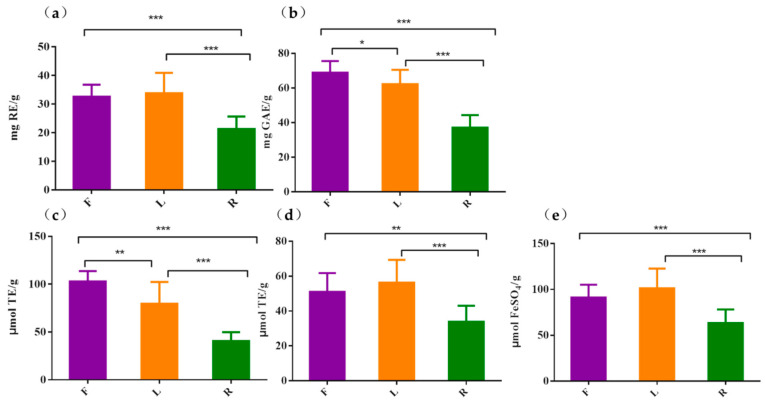
Total flavonoid contents (**a**), total phenolic contents (**b**) DPPH (**c**), ABTS (**d**), and FRAP(**e**) assays of flowers (**F**), leaves (**L**), and roots (**R**) of *Taraxacum mongolicum*. (mean ± SEM, *n* = 3) * *p* < 0.05, ** *p* < 0.01, *** *p* < 0.001.

**Figure 4 molecules-25-03260-f004:**
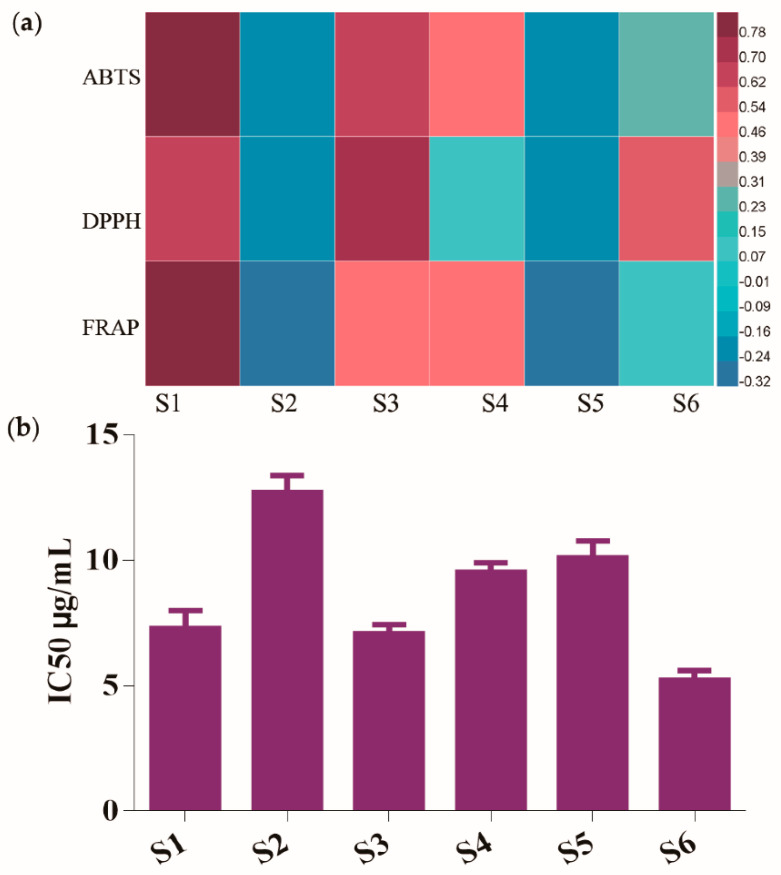
(**a**) Heatmap analysis of the Pearson correlation of the six phenolic compounds and antioxidant activities (ABTS, DPPH, FRAP). Red represents positive correlated and blue indicates negative correlated. (**b**) The IC50 values of antioxidant activity (DPPH) of six phenolic compounds. S1, caftaric acid; S2, chlorogenic acid; S3, caffeic acid; S4, cichoric acid; S5, 3,5-di-*O*-caffeoylquinic acid; S6, luteolin.

**Figure 5 molecules-25-03260-f005:**
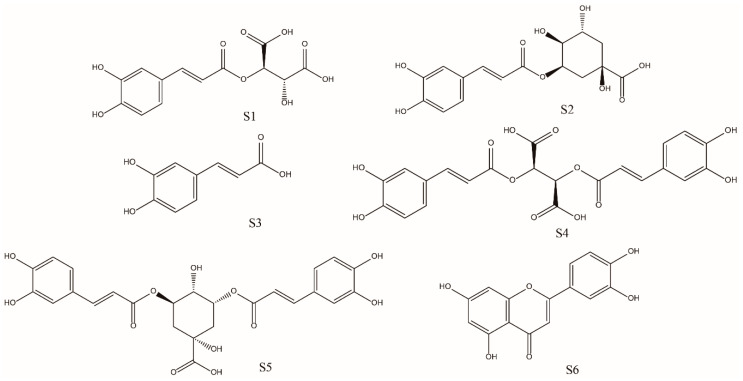
Chemical structures of the six phenolic compounds. S1, caftaric acid; S2, chlorogenic acid; S3, caffeic acid; S4, cichoric acid; S5, 3,5-di-*O*-caffeoylquinic acid; S6, luteolin.

**Table 1 molecules-25-03260-t001:** The regression equation, r^2^, linear range, limit of quantifications (LOQ), limit of detections (LOD), precision, stability, repeatability, and recovery of six analytes.

Compound	Regression Equation	r^2^	Linear Range (mg/L)	LOQ (mg/L)	LOD (mg/L)	Precision (RSD, %)		Stability (48 h) RSD (%)	Repeatability (*n* = 6) RSD (%)	Recovery (*n* = 6)	
						Intra-Day (*n* = 6)	Inter-Day (*n* = 9)			Recovery (%)	RSD (%)
Caftaric Acid	y = 33.203x + 25.550	0.9999	2.66–170	0.053	0.027	0.65	1.43	1.35	1.64	94.03	2.05
Chlorogenic Acid	**y** = 33.339x + 0.306	0.9999	0.47–60	0.188	0.043	0.90	1.28	1.28	1.76	95.89	2.30
Caffeic Acid	**y** = 53.817x − 4.734	0.9999	0.53–64	0.042	0.016	0.97	1.68	1.76	0.45	93.76	1.88
Cichoric Acid	**y** = 43.518x − 33.429	0.9999	3.39–435	0.047	0.024	0.51	0.59	0.59	0.94	108.73	2.97
3,5-di-*O*-Caffeoylquinic Acid	**y** = 38.534x − 1.189	0.9998	0.69–88	0.055	0.028	1.22	1.04	1.04	1.11	99.63	2.79
Luteolin	**y** = 49.205x + 21.672	0.9999	0.22–225	0.088	0.022	0.66	1.56	1.56	1.57	104.65	2.89

**Table 2 molecules-25-03260-t002:** The contents of six phenolic compounds in flowers (F1–F12), leaves (L1–L11) and roots (R1–R9) of *Taraxacum mongolicum* (mg/g, *n* = 3).

NO.	Caftaric Acid	Chlorogenic Acid	Caffeic Acid	Cichoric Acid	3,5-di-*O*-Caffeoylquinic Acid	Luteolin
F1	0.775	0.15	0.619	1.494	0.345	0.626
F2	0.926	0.466	0.423	4.826	0.736	0.716
F3	1.126	0.376	0.524	2.799	0.496	0.683
F4	0.698	0.339	0.229	3.803	0.456	1.308
F5	0.675	0.321	0.257	3.301	0.444	0.944
F6	1.073	0.365	0.535	2.363	0.582	0.476
F7	1.101	0.311	0.592	2.049	0.449	2.762
F8	1.399	0.414	0.738	2.934	0.611	2.506
F9	0.557	0.295	0.218	2.746	0.499	1.373
F10	1.056	0.567	0.533	2.507	1.235	0.412
F11	0.997	0.506	0.541	2.524	0.966	0.549
F12	0.745	0.368	0.261	3.94	0.449	1.006
L1	1.051	0.085	0.37	1.9	0.087	0.09
L2	1.798	0.397	0.312	9.337	0.203	0.028
L3	1.854	0.392	0.356	4.301	0.264	0.027
L4	1.242	0.271	0.248	5.191	0.191	0.097
L5	1.158	0.279	0.208	5.295	0.168	0.067
L6	1.021	0.264	0.16	4.393	0.156	0.166
L7	1.746	0.336	0.366	4.462	0.302	0.036
L8	1.537	0.095	0.37	2.359	0.063	0.396
L9	0.877	0.081	0.182	2.812	0.085	0.076
L10	1.58	0.287	0.342	4.183	0.289	0.045
L11	1.543	0.228	0.298	3.52	0.302	0.046
R1	0.398	0.358	0.086	4.928	0.49	- ^1^
R2	0.418	0.404	0.143	4.75	1.137	-
R3	0.207	0.549	0.073	2.624	0.734	0.014
R4	0.292	0.734	0.092	3.566	0.881	-
R5	0.443	0.445	0.149	5.012	0.893	-
R6	0.156	0.509	0.06	1.896	0.684	-
R7	0.397	0.42	0.144	4.383	0.9	-
R8	0.296	0.4	0.11	3.649	0.707	-
R9	0.18	0.597	0.054	2.367	0.754	-

^1^ Not detected

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
