# Peer review of "Comparison of Bioactive Phenolic Compounds and Antioxidant Activities of Different Parts of Taraxacum mongolicum"

_molecules, 2020, doi:10.3390/molecules25143260_

Round 1

Reviewer 1 Report

The manuscript deals with the chemical characterization of Taraxacum mongolicum extracts from different parts of the plant material, chromatographic analysis allowed to elucidate the main phenolic acids in each tissue, and also to establish the quantitative differences between them. Manuscript is well presented and supported bibliographically. Pearson correlation and chemometric analysis allowed to establish the effect of main phenolic compounds in the antioxidant activity of extracts. As authors stated, this manuscript may be a helpful reference for determining the strategy to follow when choosing the type of tissue in a plant material, when using as source of functional components, especially those with antioxidant activity. It is recommended to accept it in its actual state.

There is only one query for authors: In line 98 it is stated “The recoveries of the six compounds were in the range of 93.76%-108.73% with the RSD less than 2.97%.”    Why dis authors account for recoveries higher than 100%? Are these values relative contents?

Author Response

Response to Reviewer 1 Comments

The manuscript deals with the chemical characterization of Taraxacum mongolicum extracts from different parts of the plant material, chromatographic analysis allowed to elucidate the main phenolic acids in each tissue, and also to establish the quantitative differences between them. Manuscript is well presented and supported bibliographically. Pearson correlation and chemometric analysis allowed to establish the effect of main phenolic compounds in the antioxidant activity of extracts. As authors stated, this manuscript may be a helpful reference for determining the strategy to follow when choosing the type of tissue in a plant material, when using as source of functional components, especially those with antioxidant activity. It is recommended to accept it in its actual state.

Point 1: There is only one query for authors: In line 98 it is stated “The recoveries of the six compounds were in the range of 93.76%-108.73% with the RSD less than 2.97%.”    Why dis authors account for recoveries higher than 100%? Are these values relative contents?

Response 1As described in “3.5. Method validation of HPLC Analysis”, recovery of analytes was carried out for accuracy evaluation of the method. A certain amount of the six standards were added into a 0.25 g powder of six batches of the same sample, and then extracted and analyzed in sextuplicate with the same procedures. The average recoveries were estimated by the formula:

Recovery (%) = (Detected amount – Original) / Spiked amount × 100

Hence, if the recoveries of analytes were in the range of 90%-110% with the RSD less than 3%, the accuracy of the established method is acceptable.

Reviewer 2 Report

The manuscript entitled “Comparison of Bioactive Phenolic Compounds and Antioxidant Activities of Different Parts of Taraxacum mongolicum”, by authors Li Duan, Chenmeng Zhang, Yang Zhao, Yanzhong Chang and Long Guo, brings new and interesting informations about the bioactive phenolic compounds content and antioxidant potential of different parts of Taraxacum mongolicum plant. 

The authors use appropriate and complementary methods to determine the bioactive phenolic compounds in flowers, leaves, and roots of Taraxacum mongolicum (HPLC, chemometrics analysis) and their antioxidant potential (three assays for antioxidant activity evaluation, statistical analysis to establish correlation between phenolic compound content and antioxidant potential). The data show that the bioactive composition of different parts of Taraxacum mongolicum are comparable, while the relative content of the compounds were different. The components that discriminate between the different parts of plant are caftaric acid, caffeic acid and luteolin.

The measurement of antioxidant capacity of different parts of Taraxacum mongolicum show that flowers and leaves have comparable effects and greater than that of roots, positively correlated with their content of total phenols and flavonoids. The statistical analysis indicate the caftaric acid and caffeic acid as the most relevant antioxidants in Taraxacum mongolicum.

The assays and methodologies used are adequate to achieve the answer to the scientific question raised. The article is generally well written, the results are relatively well integrated in the literature, the major findings are novel, and the statistical analysis is appropriate.

There are minor aspects to be improved:

  1. At lines 139, 149 and 155 the figure numbers should be corrected in the text (3a, 3b and 3c replaced by 2a, 2b and 2c).
  2. It is not clear whether the antioxidant activities are expressed by comparing to a control without TM extract or just comparison between antioxidant activities of the three parts of plant; please specify in the Materials and Methods.
  3. Due to the placement the Materials and Methods section after the Results section, some abbreviations appear in the Results without being explained (e.g. mg RE/g or mg GAE/g for rutin equivalent or gallic acid equivalents).

Author Response

Response to Reviewer 2 Comments

The manuscript entitled “Comparison of Bioactive Phenolic Compounds and Antioxidant Activities of Different Parts of Taraxacum mongolicum”, by authors Li Duan, Chenmeng Zhang, Yang Zhao, Yanzhong Chang and Long Guo, brings new and interesting informations about the bioactive phenolic compounds content and antioxidant potential of different parts of Taraxacum mongolicum plant. 

The authors use appropriate and complementary methods to determine the bioactive phenolic compounds in flowers, leaves, and roots of Taraxacum mongolicum (HPLC, chemometrics analysis) and their antioxidant potential (three assays for antioxidant activity evaluation, statistical analysis to establish correlation between phenolic compound content and antioxidant potential). The data show that the bioactive composition of different parts of Taraxacum mongolicum are comparable, while the relative content of the compounds were different. The components that discriminate between the different parts of plant are caftaric acid, caffeic acid and luteolin.

The measurement of antioxidant capacity of different parts of Taraxacum mongolicum show that flowers and leaves have comparable effects and greater than that of roots, positively correlated with their content of total phenols and flavonoids. The statistical analysis indicate the caftaric acid and caffeic acid as the most relevant antioxidants in Taraxacum mongolicum.

The assays and methodologies used are adequate to achieve the answer to the scientific question raised. The article is generally well written, the results are relatively well integrated in the literature, the major findings are novel, and the statistical analysis is appropriate.

There are minor aspects to be improved:

Point 1: At lines 139, 149 and 155 the figure numbers should be corrected in the text (3a, 3b and 3c replaced by 2a, 2b and 2c).

Response 1Sorry for the mistakes. The figure numbers have been corrected in the revised manuscript.

Point 2: It is not clear whether the antioxidant activities are expressed by comparing to a control without TM extract or just comparison between antioxidant activities of the three parts of plant; please specify in the Materials and Methods.

Response 2Thanks for the suggestion. The antioxidant activities of flowers, leaves, roots of TM extract (60% methanol) were evaluated and compared by three antioxidant assays (DPPH, ABTS, FRAP).For DPPH and ABTS assays, the antioxidant capacities of different parts of TM could be measured in comparison to Trolox as standard, and the results were expressed as μM Trolox equivalents per gram dry weight of sample (μM TE/g). For FRAP assay, the antioxidant capacities of different parts of TM could be measured in comparison to FeSO4 as standard, and the results were expressed as μM FeSO4 equivalent per gram dry weight of sample (μM FeSO4/g). The above have been added in “3.8. Antioxidant Assays” in the revised manuscript.

Point 3: Due to the placement the Materials and Methods section after the Results section, some abbreviations appear in the Results without being explained (e.g. mg RE/g or mg GAE/g for rutin equivalent or gallic acid equivalents).

Response 3Sorry for the mistakes. The abbreviations of mg RE/g and mg GAE/g have been explained as milligrams of rutin equivalent per gram dry weight of sample and milligrams of gallic acid equivalents per gram dry weight of sample in the revised manuscript, respectively.

Reviewer 3 Report

Overall Considerations

The manuscript depicts the performance of different analytical assays of Taraxacum mongolicum in order to evaluate the differences in antioxidant content in different plant parts. A quick revision on the punctuation of sentences is needed. The manuscript is written in good English. However, some corrections regarding the data are needed.

The calibration curves must be revised or recalculated. As written further in the specific considerations, the linear correlations cannot be exactly 1 for analytical data.

The pattern and classification methods displayed in Figure 2 must be better explained; the classification and the data variables used in the calculations is not clear.

The article successfully displays the correlation between phenolic compounds and antioxidant behaviour and the separation of content by plant part is well displayed by the mathematical analysis. With the proper corrections the article could be considered for publication with revision.

 Specific Considerations

Abstract

 It would be good to reformulate some parts of the text that are sentences repeated "flowers, leaves, and roots) of Taraxacum mongolicum" (Lines 19, 22 e 23).   Line 22: Chemometrics is a wide term, it could be narrowed down to pattern exploration and classification analysis.

Line 24: “The results indicated that the bioactive compositions of different 24 parts of Taraxacum mongolicum were similar, while their relative contents were different.” How so? The statement is unclear

Line 29: “good antioxidant properties” should be defined, it is unclear to what is considered good

Line 29-31: What is the unit used to evaluate the “antioxidant capacities”

Introduction

The last paragraph is describing the methodology of the work, it should be in materials and methods, not in introduction.   Results and discussion: There was a lack of comparisons of the results found at work with other studies. This needs to be included in the entire discussion.   Materials and methods: The abts reading was monitored at an absorbance of 414 nm. This reading must be at the absorbance of 734 nm. In addition, both articles used to justify the chosen methodology also use an absorbance of 734 nm. The same occurs for dpph, the work uses an absorbance of 519 nm, while the articles chosen to justify the methodology use 517 nm. They need to be checked and corrected.

Results and Discussions

There was a lack of comparisons of the results found at work with other studies. This needs to be included in the entire discussion.  

Table 1: The authors should revise the calibration curves of the chlorogenic acid, caffeic acid, cichoric acid and luteolin. The linear coefficient (r²) for two different variables cannot be 1 unless the values are perfectly matched which is never the case for any instrumental analysis, which is never completely error-free. A linear coefficient of 1 implies that the analysis had 0 residual value and 0 deviation, moreover your regression equation wouldn’t be possible to calculate. This error may occur depending on the choice for y and x variables or round ups. Revision needed.

Line 134: “Although the bioactive phenolic constituents in different parts of TM were similar, the content 134 of each compound varied greatly.” The phrase is unclear

Line 143-152: Refrain from stabilishing PCA and PLS as purely chemometric methods. These approaches are only chemometric methods when applied to chemical data. They are mathematical models applied to a wide variety of uses. The correct denomination is “PCA and PLS-DA are mathematical approaches that can be applied to chemical or biological data, in order to recognise pattern and classify the samples”

Figure 2: What are the color labels referring too? The classification from PLS-DA or the original class assignment? This information is needed

Material and Methods

The abts reading was monitored at an absorbance of 414 nm. This reading must be at the absorbance of 734 nm. In addition, both articles used to justify the chosen methodology also use an absorbance of 734 nm. The same occurs for dpph, the work uses an absorbance of 519 nm, while the articles chosen to justify the methodology use 517 nm. They need to be checked and corrected.

Line 326-330 “Data Analysis”: Despite the good separation found in the figures displaying HCA, PCA and PLS-DA in the Results and Discussion section, the variables and the data used in the performance is missing from the methods section. Clearly inform what the data matrix used in this calculation is, along with the used variables.

Author Response

Response to Reviewer 3 Comments

Overall Considerations

The manuscript depicts the performance of different analytical assays of Taraxacum mongolicumin order to evaluate the differences in antioxidant content in different plant parts. A quick revision on the punctuation of sentences is needed. The manuscript is written in good English. However, some corrections regarding the data are needed.

Point 1: The calibration curves must be revised or recalculated. As written further in the specific considerations, the linear correlations cannot be exactly 1 for analytical data.

Response 1Thanks for the advice. The calibration curves of six analytes have been recalculated. The linear correlation (r2) of caftaric acid, chlorogenic acid, caffeic acid, cichoric acid, 3,5-di-O-caffeoylquinic acid and luteolin were 0.9999, 0.9999, 0.9999, 0.9999, 0.9998 and 0.9999, respectively. The linear correlations of the six analytes have been corrected in the revised manuscript.

Point 2: The pattern and classification methods displayed in Figure 2 must be better explained; the classification and the data variables used in the calculations is not clear.

Response 2Thanks for the advice. The HCA, PCA and PLS-DA used in the present study were described in more detail. Firstly, HCA was performed using Ward's method as the cluster method. Then, unsupervised PCA was carried out to visualize the classification trends of TM samples based on the contents of the six phenolic compounds. The first and second principal components described 59.4% and 27.1% of the variability in the original observations, and the first two principal components accounted for 86.5% of total variance. Finally, supervised PLS-DA was further employed based on the contents of the six phenolic compounds. Similar to the PCA result, the PLS-DA scores plot showed that flowers, leaves and roots of TM samples could also be readily classified into three groups with the R2Y = 0.763 and Q2 = 0.708, revealing a good classification and prediction ability of the model.

 Specific Considerations

Abstract

Point 3: It would be good to reformulate some parts of the text that are sentences repeated "flowers, leaves, and roots) of Taraxacum mongolicum" (Lines 19, 22 e 23).   

Response 3Thanks for the suggestion. These sentences have been revised as “In the present study, the bioactive phenolic chemical profiles and antioxidant activities of flowers, leaves, and roots of Taraxacum mongolicum were investigated. Firstly, a high performance liquid chromatography method combined with segmental monitoring strategy was employed to simultaneously determine six bioactive phenolic compounds in Taraxacum mongolicum samples. Moreover, chemometrics analysis was performed to compare and discriminate different parts of Taraxacum mongolicum based on the quantitative data.”

Point 4: Line 22: Chemometrics is a wide term, it could be narrowed down to pattern exploration and classification analysis.

Response 4The sentence has revised as “Moreover, multivariate statistical analysis, including hierarchical clustering analysis, principal component analysis and partial least squares discriminant analysis were performed to compare and discriminate different parts of Taraxacum mongolicum based on the quantitative data.”

Point 5: Line 24: “The results indicated that the bioactive compositions of different parts of Taraxacum mongolicum were similar, while their relative contents were different.” How so? The statement is unclear

Response 5The sentence has been deleted in the revised manuscript.

Point 6: Line 29: “good antioxidant properties” should be defined, it is unclear to what is considered good

Response 6Thanks for the advice. “good antioxidant properties” has been revised as “antioxidant properties” in the revised manuscript.

Point 7: Line 29-31: What is the unit used to evaluate the “antioxidant capacities”

Response 7The antioxidant capacities of flowers, leaves, roots of TM were evaluated and compared by three antioxidant assays (DPPH, ABTS, FRAP). For DPPH and ABTS assays, the antioxidant capacities could be measured in comparison to Trolox as standard, and the results were expressed as μM Trolox equivalents per gram dry weight of sample (μM TE/g). For FRAP assay, the antioxidant capacities could be measured in comparison to FeSO4 as standard, and the results were expressed as μM FeSO4 equivalent per gram dry weight of sample (μM FeSO4/g).

Introduction

Point 8: The last paragraph is describing the methodology of the work, it should be in materials and methods, not in introduction.  

Response 8The last paragraph introduced the purpose and research contents in this paper. It is a brief account of this research, so we think should be retained in the introduction.

Point 9: Materials and methods: The abts reading was monitored at an absorbance of 414 nm. This reading must be at the absorbance of 734 nm. In addition, both articles used to justify the chosen methodology also use an absorbance of 734 nm. The same occurs for dpph, the work uses an absorbance of 519 nm, while the articles chosen to justify the methodology use 517 nm. They need to be checked and corrected.

Response 9The ABTS radical scavenging activity of different parts of TM was performed by a commercial kit. According to the instruction manual, ABTS oxidizes to green ABTS· + with proper antioxidant and the absorbance of ABTS· + could be measured at 414 nm or 734nm. In this present study, we used 414 nm as the detected wavelength.

When the maximum absorption wavelength of the same solution is determined by different types of UV-Vis spectrophotometers, the maximum absorption wavelength is not necessarily the same. However, if the maximum wavelength is within the allowable range, this result is acceptable. Thus, the absorbance was monitored at 519 nm in DPPH assay.

Results and Discussions serveral

Point 10: There was a lack of comparisons of the results found at work with other studies. This needs to be included in the entire discussion.  

Response 10Thanks for the suggestion. Several discussions have been added in the revised manuscript.

“It has been widely reported that bioactive compounds of TM were phenolics and flavonoids. Up to now, several studies have been investigated the bioactive phenolics constitutes in whole herb of TM, but no research has been conducted to compare the contents of phenolic components in different parts of TM [16]. In the present work, the contents of the six phenolic compounds in flowers, leaves and roots of TM have been determined and the results showed that the different parts (flowers, leaves and roots) of TM showed similar chemical profiles, but the content levels of the bioactive compounds varied significantly.”

“Phenolic and flavonoid compounds were usually considered as the basis of antioxidant activities due to their hydroxyl group [20,21]. Similarly, our results indicated that the antioxidant activities of different parts of TM samples revealed a positive relationship with their total phenolic and flavonoid contents. The flowers and leavesof TM had higher contents of phenolic and flavonoid, they also showed better antioxidant activities.”

“The preliminary studies showed that the TM exhibited high antioxidant activity, but the potential antioxidant ingredients were still unknown [22].”

Point 11: Table 1: The authors should revise the calibration curves of the chlorogenic acid, caffeic acid, cichoric acid and luteolin. The linear coefficient (r²) for two different variables cannot be 1 unless the values are perfectly matched which is never the case for any instrumental analysis, which is never completely error-free. A linear coefficient of 1 implies that the analysis had 0 residual value and 0 deviation, moreover your regression equation wouldn’t be possible to calculate. This error may occur depending on the choice for y and x variables or round ups. Revision needed.

Response 11Thanks for the advice. The calibration curves of six analytes have been recalculated. The linear correlation (r2) of caftaric acid, chlorogenic acid, caffeic acid, cichoric acid, 3,5-di-O-caffeoylquinic acid and luteolin were 0.9999, 0.9999, 0.9999, 0.9998, 0.9999 and 0.9999, respectively. The linear correlations of the six analytes have been corrected in the revised manuscript.

Point 12: Line 134: “Although the bioactive phenolic constituents in different parts of TM were similar, the content of each compound varied greatly.” The phrase is unclear

Response 12The sentence has been deleted in the revised manuscript.

Point 13: Line 143-152: Refrain from stabilishing PCA and PLS as purely chemometric methods. These approaches are only chemometric methods when applied to chemical data. They are mathematical models applied to a wide variety of uses. The correct denomination is “PCA and PLS-DA are mathematical approaches that can be applied to chemical or biological data, in order to recognise pattern and classify the samples”

Response 13Thanks for the suggestion. This paragraph has been revised as “PCA and PLS-DA are mathematical approaches that can be applied to chemical or biological data, in order to recognise pattern and classify the samples, and the two chemometrics methods have been widely used in discriminating and comparing the composition of herbal medicines”.

Point 14: Figure 2: What are the color labels referring too? The classification from PLS-DA or the original class assignment? This information is needed

Response 14The different color labels represent the different parts of TM samples, Green circles represent Taraxacum mongolicum flowers, blue squares represent Taraxacum mongolicum leaves, and red triangles represent Taraxacum mongolicum roots. The information have been added in legend of Figure 2 in the revised manuscript.

Material and Methods

Point 15: The abts reading was monitored at an absorbance of 414 nm. This reading must be at the absorbance of 734 nm. In addition, both articles used to justify the chosen methodology also use an absorbance of 734 nm. The same occurs for dpph, the work uses an absorbance of 519 nm, while the articles chosen to justify the methodology use 517 nm. They need to be checked and corrected.

Response 15The ABTS radical scavenging activity of different parts of TM was performed by a commercial kit. According to the instruction manual, ABTS oxidizes to green ABTS· + with proper antioxidant and the absorbance of ABTS· + could be measured at 414 nm or 734nm. In this present study, we used 414 nm as the detected wavelength. For DPPH assay, the absorbance was monitored at 517 nm.

When the maximum absorption wavelength of the same solution is determined by different types of UV-Vis spectrophotometers, the maximum absorption wavelength is not necessarily the same. However, if the maximum wavelength is within the allowable range, this result is acceptable. Thus, the absorbance was monitored at 519 nm in DPPH assay.

Point 16: Line 326-330 “Data Analysis”: Despite the good separation found in the figures displaying HCA, PCA and PLS-DA in the Results and Discussion section, the variables and the data used in the performance is missing from the methods section. Clearly inform what the data matrix used in this calculation is, along with the used variables.

Response 16Thanks for the advice. The contents of the six phenolic compounds in flowers, leaves and roots of TM samples were used as input data for HCA, PCA and PLS-DA. This sentence has been added in “3.9. Data Analysis” in the revised manuscript.

Round 2

Reviewer 3 Report

The paper was improved, therefore in my opinion it can be accepted